# A retrospective clinical, multi-center cross-sectional study to assess the severity and sequela of Noma/Cancrum oris in Ethiopia

**Heron Gezahegn Gebretsadik**  [1]*, **Laurent Cleenewerck de Kiev** [2]

**1** School of Global Health & Bioethics, Euclid University, Banjul, Gambia, **2** EUCLID (Euclid University), Bangui, Central African Republic

* gezahegn.heron@gmail.com

## Abstract

**Data Availability Statement:** All relevant data are within the manuscript and its Supporting Information files.

### Introduction

Noma is a disfiguring gangrenous disease of the orofacial tissue and predominantly affects malnourished children. The tissue gangrene or necrosis starts in the mouth and eventually spreads intra-orally with the destruction of soft and hard tissues. If not controlled, the natural course of the condition leads to a perforation through the skin of the face, creating a severe cosmetic and functional defect, which often affects the mid-facial structures. Furthermore, the course of the disease is fulminating, and without timely intervention, it is fatal.

### Materials and methods

A retrospective clinical cross-sectional study was conducted to assess the sequela and severity of Noma in Ethiopia. Medical records of patients diagnosed with Noma were reviewed. The medical files were obtained from Yekatik 12 Hospital, Facing Africa, and the Harar project,—the three major Noma treatment centers in Ethiopia. The severity of facial tissue damage and the extent of mouth trismus (ankylosis) were examined based on the NOIPTUS score.

### Results

A total of 163 medical records were reviewed. Of those, 52% (n = 85) and 48% (n = 78) have reported left-sided and right-sided facial defects, respectively. The facial defects ranged from minor to severe tissue damage. In other words, 42.3% (n = 69), 30.7% (n = 50), 19% (n = 31), and 8% (n = 13) have reported Grade-2 (25–50%), Grade-3 (50–75%), Grade-1 (0–25%), and Grade-4 (75–100%) tissue damages respectively. Cheek, upper lip, lower lip, nose, hard palate, maxilla, oral commissure, zygoma, infra-orbital region, mandible, and chin are oftentimes the major facial anatomic regions affected by the disease in the individuals identified in our review. Complete loss of upper lip, lower lip, and nose were also identified as a sequela of Noma.

**Funding:** The author(s) received no specific funding for this work.

**Competing interests:** The authors have declared that no competing interests exist.

## Discussion

The mortality rate of Noma is reported to vary between 85% and 90%. The few survivors suffer from disfigurement and functional impairment affecting speech, breathing, mastication, and/or even leading to changes in vision. Often, the aesthetic damage becomes a source of stigma, leading to isolation from society, as well as one's family. Similarly, our review found a high level of facial tissue damage and psychiatric morbidity.

### Author summary

Noma is orofacial gangrene that rapidly disintegrates the hard and soft tissue of the face. The mortality rate of Noma varies between 85–90%. The remaining 10–15% of Noma survivors permanently suffer from severe facial deformities. Noma is a widely neglected disease affecting poor people globally. Most cases of Noma are reported from the so-called Noma belt, located south of the Sahara and runs across Africa from Senegal to Ethiopia. Though Ethiopia is one of the countries in the Noma-belt region where people, particularly children, are significantly affected by the disease, the attention given to this devastating condition is remained to be very low. Therefore, we believe that assessing the severity and sequela of Noma in Ethiopia is crucially essential to lay bare the burden of the disease and increase the overall understanding of the condition among different stakeholders. We are also convinced that the study's findings can serve as baseline data for further in-depth scientific investigations and preventive policy development.

## Introduction

Noma is a necrotizing and disfiguring condition of the orofacial and para-oral structures [1]. In many cases, the disease begins to develop from acute necrotizing ulcerative gingivitis (ANUG) [2]. ANUG is a non-contagious anaerobic infection associated with the proliferation of borrelia vincentii and fusiform bacteria [3,4]. The condition primarily affects children under the age of 10 years [3]. The major risk factors of ANUG/Noma in children are malnutrition, diarrheal diseases, measles infection, lack of proper sanitation, and poor living conditions. Alternatively, typical predisposing risk factors in young adults, including those who serve in the military, include: poor oral hygiene, smoking, viral respiratory infections, and immune defects, such as HIV/AIDS [5–7]. Characteristic features of ANUG include profuse gingival bleeding, severe soreness from gingival ulceration, halitosis (bad breath), and changes in taste. Malaise, fever, and cervical node enlargement are less commonly reported in those with ANUG. Other characteristics of ANUG include ulceration of the gingival papillae, pain, and sometimes the presence of grey pseudomembranes [8].

If left untreated, Noma often rapidly progresses to gangrenous stomatitis and gingival ulcer [9]. Within days, it spreads to adjacent hard and soft tissues by disrupting anatomical barriers, causing lysis and necrosis of bones and muscles of the orofacial region. Noma disfigures the cheek (maxilla & mandible), the floor of the mouth, head and neck, the infra-orbital region, and the nose [10]. In most instances, the lesion (wound) area is well defined (cone-shaped), with unilateral, yellowish, and necrosis and foul-smelling purulent discharge. Fetid odor, significant pain, fever, malaise, tachycardia, tachypnea, anemia, leucocytosis, and regional lymphadenopathy are common clinical findings or frequently occur (as blending symptoms with

physical exam findings). Additional lesions also may occur in distant sites, such as the scalp, neck, ear, shoulders, chest, perineum, and vulva [11].

The disease is associated with very high morbidity and mortality [9]. Associated septicemia, intracranial infection, and pneumonia are the leading causes of death. Most survivors present with prominent facial deformities, trismus or ankylosis of the temporomandibular joint, and extensive muscle and skin contracture, which leads to the difficulty of opening and closing of the mouth, thereby, trouble in mastication and swallowing, oral incontinence, and speech difficulties [12]. The problems in mastication and swallowing can also further exacerbate or cause malnutrition and as a result, many young patients experience significant stunting of growth [13]. In those with more advanced Noma, the lesions and contractures often lead to growth disturbance and result in further facial disfigurement and functional impairment [14].

In general, the disease begins as an ulcer of the mucous membrane in the mouth, which causes an edematous face; the condition extends from within out; it rapidly destroys the soft and hard tissues of the face [15]. The acute phase of the disease affects the mouth with denudation of bone, spontaneous exfoliation of teeth, necrotizing fasciitis, and lips and cheeks myonecrosis [16]. The acute phase of the disease has a high mortality, and the evolution is exceptionally rapid, causing the loss of soft and bony facial tissues within days [17,18].

Treatment of acute Noma includes transfusion of blood and intravenous fluids, administration of antibiotics, a high protein diet supplement, and debridement of necrotic areas [19]. The surgical phase is usually initiated 6 to 18 months after a period of quiescence [20]. Even after modern and sophisticated interventions (particularly reconstructive and plastic surgeries), Noma is associated with considerable morbidity and mortality, and, understandably, psychosocial impacts on the patients remain substantial [21]. Often, those patients that survive are not only severely disfigured, but also rejected from family and society [22]. However, these longer-term effects, including psychosocial aspects, are incompletely understood and an area for further study [23].

Ethiopia is one of the countries in the Noma-belt region where people, particularly children, are profoundly affected by the disease. However, attention and resources dedicated to this devastating condition remains inadequate [24]. This study was initiated to assess the sequela and severity of Noma in Ethiopia. The findings of this study can be used as a baseline for further investigation and help to fill the current knowledge gap.

## Materials and methods

### Ethics statement

The study was approved by the Addis Ababa Health Bureau Institutional Review Board (IRB), Ethical clearance committee. The approval number of the clearance statement is "A/A/H/B/ 2116/227". Medical information was kept confidential. Moreover, the researcher used patients' initials in the MCRFs and excel sheet. However, as the study was retrospective, formal consent was not sought and obtained.

### Study design

A descriptive retrospective clinical cross-sectional study was carried out to assess the sequela and severity of Noma/Cancrum oris in Ethiopia. This retrospective descriptive cross-sectional study was undertaken to provide a snapshot of the severity of Noma in Ethiopia. The sequela and severity of the disease were investigated based on written and graphic data retrieved from Yekatit 12 Hospital, Facing Africa Ethiopia, and Harar Project Ethiopia,—the three major Noma care centers in the country.

### Study location and population

The study was conducted in Addis Ababa, the capital city of Ethiopia. Patients who were diagnosed with Noma and admitted to these three treatment centers between March 2004 and December 2020 were retrospectively studied.

### Data collection and analysis

The medical records of patients with Noma were reviewed to extract relevant clinical information that help to answer the research questions. A modified case report form (MCRF) consisting of demographic and clinical information was used to collect the relevant clinical information (S1 Appendix). The demographic section of the MCRF contains the name, gender, age, physical address, telephone address, and year of admission of the patients. Whereas, the clinical part of the MCRFs primarily subdues the localization of Noma-induced anatomical lesions and associated functional limitations. The localization of lesions was based on the WHO classification, which distinguishes commissure, cheek, lip, and complex lesions. The patient's data was summarized in the MCRFs. The clinical data obtained from MCRFs was considered for data analysis. The researcher verified the validity of the CRFs. Therefore, the sequela and severity of Noma in Ethiopia were evaluated based on the relevant clinical data recorded on the MCRFs and the NOIPTUS score. The extent of jaw constrictions (ankylosis) was described in terms of NOIPTUS mouth trismus classification. According to NOIPTUS measurement, Grade-0 implies no mouth opening difficulty. Grade-1, Grade-2, Grade-3, Grade-4 imply mouth opening of $> 40mm$, 20-40mm, 1-20mm, and locked jaw respectively. Furthermore, the severity of the lesions was recorded using the NOIPTUS guideline of classifying facial tissue damage. SPSS software program was used to calculate the numerical values that are used to describe the severity of the disease.

## Results

A total of 163 patients' medical records were selected for data analysis. All the retrospectively studied participants had at least a single Noma-induced facial defect. The severity of the facial defects varies among the retrospectively studied population. The facial anatomical landmarks damaged by the disease, the severity of the tissue damage, and associated functional limitations are presented as follows.

### Left-sided facial anatomical defects

Table 1 shows that out of the 163 medical records reviewed to identify the anatomical landmarks affected with the disease, 85 of them have reported left-sided facial defects. In general, the left-sided facial anatomical regions can be grossly classified as combined and non-combined defects. Of these 85 medical records which have reported left-sided facial deformities, 30.6% (n = 26) of them have reported non-combined defects such as cheek, upper lip, lower lip, nose, and oral commissure. Whereas, the remaining 69.4% (n = 59) medical records have reported combined facial defects such as cheek and nose; upper lip, cheek and nose; and upper lip, nose, zygoma, hard palate, and maxilla. The combined defects such as cheek and nose; cheek and oral commissure; cheek, nose and upper lip; and upper lip, nose, zygoma, hard palate, and maxilla are reported only at the left side of the face. The combined defects involved multiple facial anatomical regions.

Left-sided lower lip defects with n = 9 was reported most commonly followed by upper lip with n = 7. Cheek; cheek, both lips, and oral commissure; cheek and lower lip; cheek and both lips; each with n = 6 was reported to be the other most common left-sided Noma induced

**Table 1. Noma induced facial tissue damages and frequency of the defects.**

| Affected facial anatomical region/s | Frequency of the defects | | | |
|---|---|---|---|---|
| | Left-sided | Right-sided | Neither right nor left-sided (complete loss) | Total |
| Cheek | 19 | 5 | - | 23 |
| Lower lip | 10 | 9 | - | 19 |
| Upper lip | 10 | 6 | - | 16 |
| Cheek, lower & upper lips | 4 | 6 | - | 10 |
| Cheek, oral commissure, lower & upper lips | 5 | 6 | - | 11 |
| Cheek & lower lip | 4 | 6 | - | 10 |
| Cheek, nose, lower & upper lips | 5 | 4 | - | 9 |
| Cheek & upper lip | 5 | 4 | - | 9 |
| Lower & upper lips | 3 | 2 | - | 5 |
| Oral commissure, lower & upper lips | 4 | 2 | - | 6 |
| Nose | 3 | 3 | - | 6 |
| Nose & upper lip | 3 | 3 | - | 6 |
| Cheek, nose & upper lip | 1 | 3 | - | 4 |
| Oral commissure & upper lip | 2 | 2 | - | 4 |
| Nose & upper lip | 1 | 3 | - | 4 |
| Cheek & oral commissure | - | 3 | - | 3 |
| Oral commissure | 2 | 1 | - | 3 |
| Maxilla & upper lip | 1 | 2 | - | 3 |
| Lower lip | - | - | 2 | 2 |
| Nose | - | - | 2 | 2 |
| Central lower lip | - | - | 1 | 1 |
| Central upper lip & nose | - | - | 1 | 1 |
| Chin & upper lip | 1 | - | - | 1 |
| Cheek & nose | - | 1 | - | 1 |
| Cheek, lower lip, nose & mandible | 1 | - | - | 1 |
| Infra-orbital region, oral commissure, nose and zygoma | 1 | - | - | 1 |
| Hard palate, maxilla, nose, upper lip & zygoma | - | 1 | - | 1 |
| Total | 85 | 72 | 6 | 163 |

defects. On the other hand, upper lip, cheek, and nose; upper lip, nose, hard palate and maxilla; upper lip and maxilla; and cheek and nose, each with n = 1 were less frequently reported left-sided anatomical defects.

## Right-sided facial anatomical defects

Table 1 reveals that out of the 163 medical records reviewed to identify the anatomical landmarks affected with the condition, 72 of them have reported right-sided facial deformities. In general, the left-sided facial anatomical regions can be grossly classified as combined and non-combined defects. Out of the total 72 medical records which have reported left-sided facial deformities, 61.1% (n = 44) of them have reported non-combined defects such as cheek, upper lip, lower lip, nose, and oral commissure. Whereas, the remaining 38.9% (n = 28) medical records have reported combined facial defects such as cheek and nose; upper lip, cheek and nose; and nose, zygoma, oral commissure, maxilla, and infra-orbital region. The combined defects, such as the nose, zygoma, oral commissure, and infra-orbital, lower lip, cheek, mandible, and nose, and lower lip and chin are reported only on the right side of the face. The combined defects involved multiple facial anatomical regions.

Righted sided cheek defects with n = 19 was reported most commonly followed by the upper lip and lower lip each with n = 10. Cheek and both lips with n = 9; cheek and upper lip with n = 7; both lips, cheek, and nose with n = 5, and both lips, cheek, and oral commissure with n = 5 were reported to be the other most common right-sided Noma induced defects. On the other hand, lower lip and chin; upper lip and nose; lower lip, cheek, mandible and nose; and nose, zygoma, oral commissure, maxilla, and infra-orbital region each with n = 1 were less frequently reported right-sided anatomical defects.

## Neither left nor right-sided facial anatomical defects

Table 1 demonstrates that out of the 163 medical records reviewed to identify the anatomical landmarks affected with Noma, 6 of them have reported complete loss of facial tissues. A single entire nose and central upper lip loss, two total nose losses, two complete lower lip loss, and one complete upper lip loss. Out of the 6, only one defect was combined (nose and upper lip). The remaining five defects were non-combined. Only one of the medical records with complete nose loss has reported grade-II mouth opening. No missing teeth or limited mouth opening were reported in the remaining five medical records.

## Overall facial anatomical defects

A total of 163 medical records have been reviewed to assess the epidemiology of Noma in Ethiopia. The socio-demographic data of the cases admitted in the three Noma treatment facilities were reviewed from the medical records. All the relevant clinical information obtained from the medical records has been organized and synthesized to come up with the following findings. The sequela of the disease reported in the medical records is classified into combined and non-combined. The combined sequela involved two or more anatomical region defects. The cheek, upper lip, lower lip, nose, and oral commissure are the non-combined anatomical defects reported in the overall medical records. The non-combined anatomical abnormalities consisted of 44.2% (n = 72) of the total defects reported in the whole medical records. The cheek, the lower lip, and the upper lip are the most frequently reported non-combined defects with n = 23, n = 19, and n = 16, respectively.

The least frequently reported non-combined Noma-induced facial defects are oral commissure with n = 3 and nose with n = 6. Furthermore, there are five non-combined complete loss defects.

Nearly 56% (n = 91) of the total facial defects reported in the overall medical records are combined in form. Upper lip, lower lip, and cheek with n = 10; cheek and upper lip with n = 9; cheek and lower lip with n = 10; upper lip, lower lip, cheek and nose with n = 9; and upper and lower lip with n = 10 was the most frequently reported combined sequela of the Noma in the overall medical records.

The least commonly reported combined defects with n = 1 are cheek and nose; lower lip and chin; lower lip, cheek, mandible, and nose; nose, zygoma, oral commissure, maxilla, and infra-orbital region; and upper lip, nose, zygoma, hard palate, and maxilla. Furthermore, there is a single non-combined complete loss defect.

## The severity of the facial defects

The extent of the disease sequela was reported in the medical records based on the NOIPTUS guide of classifying facial tissue damage. As Table 2 describes, the NOIPTUS classifies facial anatomical tissue damage into four classes. The classes are graded from 1 to 4, depending on the severity of facial tissue damage. Grade-1, Grade-2, Grade-3 and Grade-4 imply 0–25%, 25–50%, 50–75%, and 75–100% facial anatomical damages respectively. All the anatomical

**Table 2. Noma cases with different levels of facial tissue damages (NOITULP Grade).**

| Number of Noma cases | Tissue damage by a percent | NOITULP Grade |
|---|---|---|
| 69 | 25–50% | Grade 2 |
| 50 | 50–75% | Grade 3 |
| 31 | 0–25% | Grade 1 |
| 13 | 75–100% | Grade 4 |
| 163 | Total number of Noma cases with facial tissue damages | |

landmarks reported in the reviewed medical records have shown minor to severe tissue damage. Cheek, upper lip, lower lip, nose, hard palate, maxilla, oral commissure, zygoma, infraorbital region, mandible, and chin are the anatomical landmarks reported to be damaged in the medical records. Accordingly, 42.3% (n = 69) of the total medical records reviewed have reported Grade-2 facial anatomical damages. Furthermore, 30.7% (n = 50), 19% (n = 31), and 8% (n = 13) have reported Grade-3, Grade-1, and Grade-4 tissue damages respectively.

## Impaired activity of the mouth

Out of the six neither left nor right-sided anatomical defects, only one of the medical records with complete nose loss has reported grade-II mouth opening. The remaining medical records have reported no difficulties with mouth opening. The impaired mouth activities that are reported in the medical records are explained in terms of NOITULP's mouth opening classification. Table 3 shows the NOITULP's mouth opening measurement scales. Grade-0 implies no mouth opening difficulty. Grade-1, Grade-2, Grade-3, Grade-4 imply mouth opening of > 40mm, 20-40mm, 1-20mm, and locked jaw respectively. Accordingly, except 23.3% (n = 38) and 4.9% (n = 8) medical records, which have reported intact mouth opening (Grade 0) and with no mouth opening information, respectively, the remaining 71.8% (n = 117) have reported grade-2 to grade-4 mouth opening limitations that related with the disease. Of these, 32.5% (n = 53), 25.2% (n = 41), and 13.5% (n = 22) have reported grade-1, grade-2, and grade-3 mouth opening limitations, respectively. Furthermore, a single locked jaw (grade-4) case has been reported.

## Dental involvement

Dental involvement has been reported in the majority of the patients in this study. Out of the total 163 patients reviewed, 7.4% (n = 12) had no dental 266 information. On the other hand, 25.8% (n = 42) are reported to have healthy teeth, while the remaining 66.9% (n = 109) have reported a varying number of missing teeth. The number of missing teeth ranges from 1 to 19 in patients affected by the disease. The reported missing teeth have been classified into four groups for the sake of understanding. Group-1, Group-2, Group-3, and Group-4 consisted of medical records that reported 1–5, 6–10, 11–15, and 16–20 missing teeth, respectively.

**Table 3. Noma cases with different levels of mouth opening limitations (NOITULP Grade).**

| Number of Noma cases | Mouth opening by millimeter | NOITULP Grade |
|---|---|---|
| 53 | > 40mm | Grade 1 |
| 41 | 20-40mm | Grade 2 |
| 22 | 1-20mm | Grade 3 |
| 1 | Locked jaw | Grade 4 |
| 117 | Total number of Noma cases with ankylosis | |

Accordingly, 43 medical records have reported 1–5 missing teeth each in group-1. Another 43 medical records have reported 6–10 missing teeth each in group 2. On the other hand, 15 medical records have reported 11–15 missing teeth each in group-3. Similarly, another eight medical records have reported 16–20 missing teeth each in group-4. Furthermore, out of the 109 medical records, 14 have reported different levels of disoriented teeth.

## Discussion

Noma is a disfiguring necrotizing condition of the orofacial tissues. The condition may be an extension of ANUG (acute necrotizing ulcerative gingivitis) [4]. In Africa, there is a high prevalence of ANUG in children that ranges from 15% to 60%, depending on the region and the degree of poverty [15,25]. Accordingly, most cases of Noma are reported from the so-called Noma belt in Africa, which is located south of the Sahara and runs across Africa from Senegal to Ethiopia [12,21,26]. Recently, isolated cases of Noma have been reported from developed countries. The burden of the disease can be explained through its high mortality rate and psychosocial morbidity [14,27]. Noma survivals are often left with devastating orofacial defects [23]. All cases reviewed in this study have reported minor to severe orofacial tissue damages. According to the NOITULP tissue damage classification system, 73% of the total cases with Noma have reported 25–75% facial tissue damage. Nearly 8% have reported 75–100% facial anatomical landmarks damage. Furthermore, a complete absence of facial tissues such as the nose, upper lip, and lower lip has been reported.

Most defects resulting from Noma involve the lateral and anterolateral aspects of the face and are often combined with severe functional deficits [19]. A subgroup, commonly called "central Noma," is composed of defects of the upper lip, maxillary soft tissues, premaxilla, nasal cartilaginous infrastructure, and soft tissues. In contrast to unilateral involvement of the face, central Noma does not affect the opening of the jaw; however, it results in severe mutilation, with disfiguring three-dimensional defects erasing any individual traits from a face [28]. Various forms of post-Noma defects were reported among 84.6% of the retrospectively studied population in Nigeria [29]. Furthermore, the estimated incidence of Noma in the north-central zone was found to be 8.3 per 100000 with a range of 4.1–17.9 per 100000 across various states. Period prevalence of Noma–which incorporated all cases seen within the study period–was also reported to be 1.6 per 100000 population at risk [30]. Among the Noma cases involved in a hospital-based retrospective study in north-western Nigeria, 84.3% had manifest outer and inner cheek layer lesions. The study examined 1923 patients admitted to the hospital from January 1999 to December 2011 [31]. Another study, which assessed the outcomes at 18 months of 37 surgically treated Noma cases at the Noma Children's Hospital, Sokoto, Nigeria revealed 36.0% of outer cheek involvement among the studied population [32]. Similarly, another research reported 26% to 50% outer cheek tissue loss [33].

Oral incompetence, speech difficulties, post-wound healing ankylosis, and dental or malalignment are reported as a sequela of Noma [34]. In this study, 74.9% (n = 122) have reported grade 1 to grade 4 levels of trismus, and 66.9% (n = 109) have reported a different number of missing teeth. Furthermore, a significant number of the cases have reported dental anarchy.

Oral incompetence, speech difficulties, airway challenges, chemotherapy-induced neutropenia, post wound healing ankylosis, and dental anarchy are reported as a sequela of Noma [35] [36] [37] [24]. Depending on the degree of inner and outer cheek lining deficit, trismus can result simply from soft tissue contracture that the cheek undergoes with resultant scarring [38]. Bony trismus can also result from the fusion of the coronoid process to the zygoma and may require surgical intervention [39] [40]. A retrospective study that investigated the long-term results of trismus release among Noma patients reported a poor prognosis. This study

showed that trismus is one of the most disabling sequelae among Noma survivals [41]. Out of the 163 medical records reviewed, 74.9% (n = 122) have reported grade 1 to grade 4 levels of trismus according to the NOITULP classification system for Noma-induced ankylosis. On the other hand, 66.9% (n = 109) have reported a different number of missing teeth. Furthermore, of these, 14 have reported varying levels of dental anarchy.

## Conclusion

Noma can only be described as a terrifying disease considered 'too disturbing' to feature in developed countries' mass media. As a result, there is little awareness of the dire need to eradicate this disease, notably in East Africa. Noma oftentimes occurs in impoverished individuals, particularly in malnourished individuals. The condition is associated with significant morbidity and mortality. The findings of this study shed light on the severity of Noma in Ethiopia. Most of the surveyed patients suffered from extensive Noma-induced facial disfigurements, which exposed them to eminent functional limitations, social discrimination, and negative psycho-social effects. This work provides baseline data for calculation of the disease burden which can explain better the disease's ill-outcomes. Whereas other health challenges such as cleft palate have received global attention and funding, Noma remains tragically unknown and neglected. While prevention and eradication of Noma should be specifically considered among the Sustainable Development Goals, the urgency of psychological and physical rehabilitation for patients with Noma should be treated as an international humanitarian emergency.

## Supporting information

**S1 Appendix. A modified case report form (MCRF) consisting of demographic and clinical information.**
(DOCX)

## Acknowledgments

I would like to acknowledge with gratitude the School of Global Health and Bioethics at EUCLID (Pôle Universitaire EUCLIDE) for supporting the project with useful scientific advice.

## Author Contributions

**Conceptualization:** Heron Gezahegn Gebretsadik, Laurent Cleenewerck de Kiev.

**Data curation:** Heron Gezahegn Gebretsadik.

**Formal analysis:** Heron Gezahegn Gebretsadik.

**Methodology:** Heron Gezahegn Gebretsadik, Laurent Cleenewerck de Kiev.

**Supervision:** Laurent Cleenewerck de Kiev.

**Validation:** Laurent Cleenewerck de Kiev.

**Writing – original draft:** Heron Gezahegn Gebretsadik.

**Writing – review & editing:** Heron Gezahegn Gebretsadik, Laurent Cleenewerck de Kiev.

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
