## [Decision Letter · Decision Letter 0]

15 May 2022

Dear Dr Gebretsadik,

Thank you very much for submitting your manuscript "A retrospective clinical, multi-center cross-sectional study to assess the severity and sequela of Noma/Cancrum oris in Ethiopia" for consideration at PLOS Neglected Tropical Diseases. As with all papers reviewed by the journal, your manuscript was reviewed by members of the editorial board and by several independent reviewers. In light of the reviews (below this email), we would like to invite the resubmission of a significantly-revised version that takes into account the reviewers' comments. 

We cannot make any decision about publication until we have seen the revised manuscript and your response to the reviewers' comments. Your revised manuscript is also likely to be sent to reviewers for further evaluation.

Sincerely,

Joseph M. Vinetz

Deputy Editor

Joseph Vinetz

Deputy Editor

Reviewer's Responses to Questions

**Key Review Criteria Required for Acceptance?**

**Methods**

-Are the objectives of the study clearly articulated with a clear testable hypothesis stated?

-Is the study design appropriate to address the stated objectives?

-Is the population clearly described and appropriate for the hypothesis being tested?

-Is the sample size sufficient to ensure adequate power to address the hypothesis being tested?

-Were correct statistical analysis used to support conclusions?

-Are there concerns about ethical or regulatory requirements being met?

Reviewer #1: 2. Methodology : Fair but no words on the software employed for analysis

Reviewer #2: (No Response)

**Results**

-Does the analysis presented match the analysis plan?

-Are the results clearly and completely presented?

-Are the figures (Tables, Images) of sufficient quality for clarity?

Reviewer #1: 3. RESULTS

Some Observation:

“All the retrospectively 138 studied study participants” a sentence like this affected the flow of the write up. There is a need to appreciate that once the study method has been mentioned, generated data should be used freely example “138 participants were studied”

I am of the opinion that a table summarizing left sided and right sided deformities will be easier to follow.

Also when using a table, its important to mention it at the beginning of the sentence rather than the end. EG Table 1 shows the distribution of facial defects on the right side of the face observed from the study population 

Ethics is always a controversial issue. However blocking the face in these subjects has substancially affected the true extent and appreciation and hence limiting how much the public need to be educated. So I am of the opinion that the face should be presented following informed consent. I appreciate the fact that it is a retrospective study but all surgeons are married to our patients and I am sure some of them will still be around.

Reviewer #2: yes - most images are good quality - 1 noticeably blurry - on the cut-off for what I would consider acceptable for an image

**Conclusions**

-Are the conclusions supported by the data presented?

-Are the limitations of analysis clearly described?

-Do the authors discuss how these data can be helpful to advance our understanding of the topic under study?

-Is public health relevance addressed?

Reviewer #1: 5. Conclusion

What was presented was not a summary. Conclusion should be the opinion arrived from discussion. The write up looks like another view. A revision is needed.

“Noma is a debilitating disease primarily affecting the poor and has been called the “face of poverty.” It occurs among the underprivileged society where patients do not have access to proper medical care. Those who do develop Noma are doomed to suffer the tragic consequences of disfigurement and functional impairment, or death.” These are established facts from past studies, so cant be a conclusion of this study

Reviewer #2: yes - writing needs to be improved - I have made the majority of required grammatical changes. This manuscript will require another proofread to make sure no errors remain.

**Editorial and Data Presentation Modifications?**

Reviewer #1: Substantial revision important

Reviewer #2: (No Response)

**Summary and General Comments**

Reviewer #1: Overall: The work of the authors is commendable. Despite the debilitating nature of cancrum oris, information about the disease is sparse and hence the study is very relevant to the advancement of knowledge on the subject. However substantial revision is needed to be acceptable as a scientific publication.

Reviewer #2: Thank you for your submission.

Detailed review.

Would make changes to the manuscript as follows:

In the abstract 

Would reword:

The facial defects depicted minor to severe tissue damages

to:

The facial defects ranged from minor to severe tissue damage.

In the abstract

The precise meaning of this sentence was unclear to me:

In general, 73% (n=119) of medical records have reported 25-75% of facial tissue damages.

in abstract would change 'chewing' to 'mastication'

in abstract would change ...or even vision to ...and/or even lead to changes in vision.

in abstract

Cheek, upper lip, lower lip, nose, hard palate, maxilla, oral commissure, zygoma, infra-orbital region, mandible, and chin are the major facial anatomic regions affected by the disease

would change to:

Cheek, upper lip, lower lip, nose, hard palate, maxilla, oral commissure, zygoma, infra-orbital region, mandible, were the major facial anatomic regions affected by the disease in the individuals identified in our review.

in abstract 

would reword 

Often, the aesthetic damage becomes a source of stigma leading to isolation from society and the family

to:

Often, the aesthetic damage becomes a source of stigma, leading to isolation from society, as well as one's family.

would reword closing few sentences of abstract (currently below)

Furthermore, patients reported high psychiatric morbidity. These facts can well describe the severity of the disease. Thus, the findings of this study also support these assertions. Three fourth of the retrospectively studied Noma cases were presented with 25-75% of facial tissue damages.

to:

Similarly, our review found a high level of psychiatric morbidity. Would then remove "

These facts can well describe the severity of the disease. Thus, the findings of this study also support these assertions"

then

this last sentence of the abstract - below - was also unclear to me (and also grammatically incorrect as written). 

this needs to be better explained

Three fourth of the retrospectively studied Noma cases were presented with 25-75% of facial tissue damages.

line 18- would not capitalize oro-facial

line 19 - consider removing :debilitated" from the sentence - it's meaning is vague in this context

lines 20 to 22:

If not controlled the condition perforates the facial skin and causes severe damage, particularly to the mid-facial structures.

would reword sentence to:

If not controlled, the natural course of the condition leads to a perforation through the skin of the face, creating a severe cosmetic and functional defect, which often affects the mid-facial structures.

line 33-34

The facial defects depicted minor to severe tissue damages.

again,

would change to below (important to remove 'depicted' as that is not the best word choice for what you are trying to describe - and the below rewording is one way to accomplish this.

to:

The facial defects ranged from minor to severe tissue damage.

line 35-36

reword - and chin are the major facial anatomic regions affected by the disease.

to:

and chin are oftentimes the major facial anatomic regions affected by Noma in general. 

then - the next line allows you to describe your findings. 

line 36-37

This sentence needs to be reworded (I mentioned this earlier - but the precise meaning of the sentence is unclear). I would remove in general, as it is apparent that you are talking about your study individuals. 

line 41 - again change 'chewing' to mastication and 

change

...or even vision to ...and/or even lead to changes in vision.

line 49

oro-facial should not be capitalized 

line 49 - would remove the word - apparatus 

would change to:

line 49

would chnage Noma is a necrotizing and disfiguring disease of the Oro-facial apparatus (1)

to:

Noma is a necrotizing and disfiguring condition of the oro-facial and para-oral structures (1).

line 54- measle should be measles

line 54 - would remove 

debilitating disease from the sentence - as this term is vague

line 54 would change phrase 

bad living conditions to:

poor living conditions 

line 54 would change 

On

55 the other hand, the typical predisposing factors in young adults, especially in armed forces,

56 are poor oral hygiene, smoking, viral respiratory infections, and immune defects such as in

57 HIV/AIDS (5) (6) (7).

to:

Alternatively, typical predisposing risk factors in young adults, including those who serve in the military include: poor oral hygiene, smoking, viral respiratory infections, and immune defects, such as HIV/AIDS.

line 57 'characteristics' should be 'characteristic' - that is making it singular rather than plural 

line 58 would change

'and bad taste'

to: 

changes in taste

line 58 would change 

Malaise, fever, and cervical lymph node enlargements are rare but possible disease symptoms

to:

Malaise, fever, and cervical node enlargement are less commonly reported in those with ANUG.

line 60

and sometimes, greyish pseudomembranes (8).

would change to:

and sometimes the presence of grey pseudomembranes (8)

line 62 - replace 'the disease' with 'Noma'

line 67

would change:

and blackish necrosis 

to:

necrosis

line 68

would change 

increased respiratory rate

to:

tacyhpnea 

line 69

and regional lymphdenopathy are typical.would changeto:

...and regional lymphadenopathy are common clinical findings or frequently occur (as blending symptoms with physical exam findings)

line 76 would replacechewing with mastificaton 

line 77 

evidence would be changed to:

experience.

line 77 to 78 - would change the beginning of the sentence to: In those with more advanced Noma, ...

line 80 - it appearsas though are you describing the initial stages of Noma - however, this order can be improved - as I would talk above the early stages of Noma - and then go through a discussion of the progression - to allow for readersto understand the natural history of Noma, with regardsto it's temporal progression 

lines 89-90

the outcome is reported to be less than complete recovery

would change to:

Noma is associated with considerable morbidity and mortality

line 91 

Often, survival patients

change to:

Often, those patients that survive...

line 92 to 93

However, these longer-term effects and particularly

93 the psychosocial aspects have been studied rarely.

would reword to:

However, these longer-term effects, including psychosocial aspects, are incompletely understood and an area for further study.

would change:

However, the attention given to this ruinous condition is remained to be very low

to:

However, attention and resources dedicated to this devastating condition remains inadequate.

line 96-97 - would remove this sentence

So far, there are no considerable systematized studies conducted

97 in Ethiopia regarding Noma

line 98 - would change

the country

to:

Ethiopia

line 99 would change

scientific knowledge gap

to: 

current knowledge gap

line 104 change 

considered

to:

undertaken

subdues

perhaps you mean:

describes?

line 127 - put an "and" in before the word 'locked"

All the patients’

134 information was treated secretly

would rephrase to:

Medical information was kept confidential 

line 234 - would change damages to damage 

line 276 - would change 

Noma is a disfiguring necrotizing disease of the Oro-facial tissue

to:

Noma is a disfiguring necrotizing condition of the oro-facial tissues.

It is the descendant of ANUG

277 (4)

would change to:

Noma may be an extension of ANUG (acute necrotizing ulcerative gingivitis).

line 283 - I would remove this phrase phrase from the sentence

"which resembles a dramatic amalgamation of

oncologic, congenital, and traumatic deformities "

line 296

disclosed 84.3%

would reword- I believe you mean 84.3% of the cases had involvement of the cheeks

line 301 to 306 would remove the discussion of the below case report

A case report presented a 20-year-old Laotian woman

302 presented with a large facial defect and bilateral trismus. The report revealed a major soft-

303 tissue defect involving the right cheek, nasal ala, upper lip and oral commissure, and severe

304 trismus. The defect was reported to be the precursor of foul-smelling and fulminating ulcers

305 developed over the patient’s right cheek and evolved into a black eschar that eventually

line 306 orofacial should not be capitalized 

line 311 - would better define "dental anarachy" the word anarachy in this context is unclear 

line 319 - would remove "tenacious" from the sentence 

line 320 would replace

need

 with require 

line 329 - would change 

among the underprivileged society where patients do not have access to

330 proper medical care.

to:

Noma oftentimes occur in impoverished individuals, particularly in malnourished individuals.

Those who do develop Noma are doomed to suffer the tragic

331 consequences of disfigurement and functional impairment, or death

change to:

Noma is associated with significant morbidity and mortality.

line 334

change

psycho-social crisis

to: negative pyscho-social effects 

The overall burden of the disease should be

335 better explained by calculating the disability-adjusted life year (DALY). As DALY is a measure

336 of the number of years lost due to ill-health (physical, mental and social), disability, or early

337 death. 

This work can also provide the basis for burden of diseases calculations that can explain

338 better the disease ill-outcomes and generally a gateway to gradually more complex, yet also

339 increasingly improved descriptions of the reality.

would remove these above three sentences - did you provide DALYs/estimate?

PLOS authors have the option to publish the peer review history of their article (what does this mean?). If published, this will include your full peer review and any attached files.

Reviewer #1: Yes: Dr Seidu Adebayo Bello

Reviewer #2: No
---

## [Decision Letter · Decision Letter 1]

26 Jul 2022

Dear Dr. Gebretsadik,

Thank you very much for submitting your manuscript "A retrospective clinical, multi-center cross-sectional study to assess the severity and sequela of Noma/Cancrum oris in Ethiopia" for consideration at PLOS Neglected Tropical Diseases. As with all papers reviewed by the journal, your manuscript was reviewed by members of the editorial board and by several independent reviewers. The reviewers appreciated the attention to an important topic. Based on the reviews, we are likely to accept this manuscript for publication, providing that you modify the manuscript according to the review recommendations. 

Editorial comments: 

1. Table……………………………Line tables are not acceptable in scientific writing. PLs employ no line designs to eliminate the lines

2. Dental involvement has been reported in the majority of medical records reviewed in this study. Out of the total 163 medical records reviewed, 7.4% (n=12) had no dental 266 information………….In scientific reseach , source of information could be medical records also known as case file or a proforma. But once inforemation are retrieved, these sources are dropped and we focus on the subjects. The above statement should be repframed as follows:

 Dental involvement has been reported in the majority of the patients in this study. Out of the 

 total 163 patients reviewed, 7.4% (n=12) had no dental 266 information

3. Discussion: The authors have shown that they understand how to employ result for discussion, however it takes the reader 26 lines before the mention of the study results. Just a brief introduction is needed before bringing your result. Maximum within 5 to 10 lines.

Sincerely,

Joseph M. Vinetz

Section Editor

Joseph Vinetz

Section Editor

Reviewer's Responses to Questions

**Key Review Criteria Required for Acceptance?**

**Methods**

-Are the objectives of the study clearly articulated with a clear testable hypothesis stated?

-Is the study design appropriate to address the stated objectives?

-Is the population clearly described and appropriate for the hypothesis being tested?

-Is the sample size sufficient to ensure adequate power to address the hypothesis being tested?

-Were correct statistical analysis used to support conclusions?

-Are there concerns about ethical or regulatory requirements being met?

Reviewer #1: Methodology is in order

**Results**

-Does the analysis presented match the analysis plan?

-Are the results clearly and completely presented?

-Are the figures (Tables, Images) of sufficient quality for clarity?

Reviewer #1: Analysis and figures well presented. The tables should be change dfrom Line design to non line design

**Conclusions**

-Are the conclusions supported by the data presented?

-Are the limitations of analysis clearly described?

-Do the authors discuss how these data can be helpful to advance our understanding of the topic under study?

-Is public health relevance addressed?

Reviewer #1: Appropriate

**Editorial and Data Presentation Modifications?**

Reviewer #1: Its ok

**Summary and General Comments**

Reviewer #1: Substaantial revision done

PLOS authors have the option to publish the peer review history of their article (what does this mean?). If published, this will include your full peer review and any attached files.

Reviewer #1: Yes: Dr Seidu Bello

Figure Files:

Data Requirements:

Reproducibility:

References

---

## [Editor Report · Decision Letter 2]

1 Sep 2022

Dear Dr. Gebretsadik,

We are pleased to inform you that your manuscript 'A retrospective clinical, multi-center cross-sectional study to assess the severity and sequela of Noma/Cancrum oris in Ethiopia' has been provisionally accepted for publication in PLOS Neglected Tropical Diseases.

Best regards,

Joseph M. Vinetz

Section Editor

Joseph Vinetz

Section Editor

---

## [Editor Report · Acceptance letter]

8 Sep 2022

Dear Dr. Gebretsadik,

We are delighted to inform you that your manuscript, "A retrospective clinical, multi-center cross-sectional study to assess the severity and sequela of Noma/Cancrum oris in Ethiopia," has been formally accepted for publication in PLOS Neglected Tropical Diseases.

Best regards,

Shaden Kamhawi

co-Editor-in-Chief

Paul Brindley

co-Editor-in-Chief
